# Phenotypic and Genomic Characterization of ESBL- and AmpC-β-Lactamase-Producing *Enterobacterales* Isolates from Imported Healthy Reptiles

**DOI:** 10.3390/antibiotics13121230

**Published:** 2024-12-20

**Authors:** Franziska Unger, Tobias Eisenberg, Ellen Prenger-Berninghoff, Ursula Leidner, Torsten Semmler, Christa Ewers

**Affiliations:** 1Institute of Hygiene and Infectious Diseases of Animals, Faculty of Veterinary Medicine, Justus Liebig University Giessen, 35392 Giessen, Germany; franziska.unger@vetmed.uni-giessen.de (F.U.); ellen.prenger-berninghoff@vetmed.uni-giessen.de (E.P.-B.); ursula.leidner@vetmed.uni-giessen.de (U.L.); 2Hessian State Laboratory, 35392 Giessen, Germany; tobias.eisenberg@lhl.hessen.de; 3Genome Competence Centre, Robert Koch Institute, 13353 Berlin, Germany; semmlert@rki.de

**Keywords:** *Enterobacterales*, antimicrobial resistance, ESBL, AmpC-β-lactamases, whole genome sequencing

## Abstract

Background/Objectives: Reptiles are known reservoirs for members of the *Enterobacterales*. We investigated antimicrobial resistance (AMR) patterns, the diversity of extended-spectrum-/AmpC-β-lactamases (ESBL/AmpC) genes and the genomic organization of the ESBL/AmpC producers. Methods: A total of 92 shipments with 184 feces, skin, and urinate samples of live healthy reptiles were obtained during border inspections at Europe’s most important airport for animal trade and screened for AMR bacteria by culture, antimicrobial susceptibility testing, and whole genome sequencing (WGS) of selected isolates. Results: In total, 668 *Enterobacterales* isolates with phenotypic evidence for extended-spectrum-/AmpC-β-lactamases (ESBL/AmpC) were obtained, from which *Klebsiella* (*n* = 181), *Citrobacter* (*n* = 131), *Escherichia coli* (*n* = 116), *Salmonella* (*n* = 69), and *Enterobacter* (*n* = 52) represented the most common groups (other genera (*n* = 119)). Seventy-nine isolates grew also on cefotaxime agar and were confirmed as ESBL (*n* = 39) or AmpC (*n* = 39) producers based on WGS data with respective genes localized on chromosomes or plasmids. Isolates of *E. coli* contained the most diverse set of ESBL genes (*n* = 29), followed by *Klebsiella* (*n* = 9), *Citrobacter*, and *Enterobacter* (each *n* = 1). Contrarily, AmpC genes were detected in *E. coli* and *Citrobacter* (*n* = 13 each), followed by *Enterobacter* (*n* = 12) and *Klebsiella* (*n* = 4). Isolates of *Salmonella* with ESBL/AmpC genes were not found, but all genera contained a variety of additional AMR phenotypes and/or genotypes. MLST revealed 36, 13, 10, and nine different STs in *E. coli*, *Klebsiella*, *Citrobacter*, and *Enterobacter*, respectively. Conclusions: A significant fraction of the studied *Enterobacterales* isolates possessed acquired AMR genes, including some high-risk clones. All isolates were obtained from selective media and also wild-caught animals carried many AMR genes. Assignment of AMR to harvesting modes was not possible.

## 1. Introduction

Members of the *Enterobacterales* have been identified as the predominant group of intestinal microorganisms in reptiles [1]. The high frequency of *Salmonella* spp. shedding has gained especially significant attendance because of its zoonotic potential [2]. Although this issue is mainly a problem of insufficient hygiene, this debate has often been misused as an argument to generally restrict wildlife importation and reptile keeping in the recent past [3,4]. Some earlier approaches have identified MDR enterobacteria in dedicated single isolates from reptiles’ intestinal tracts by respective phenotypes or genotypes. Studies were mostly carried out in a limited set of reptile species and investigations largely brought β-lactam and fluorquinolone resistances into focus [5]. In this respect, healthy as well as debilitated sea turtles received special attention due to their global ocean habitats and as a proxy for marine ecosystem health [6,7,8,9]. Regardless, zoonotic transmission occurs mainly through direct and indirect contact via the fecal–oral route, through reptile bites [10] or by consumption, because reptiles also represent important food sources for human nutrition in many parts of the world [11]. The same transmission routes can also be anticipated for MDR bacteria, and a spillover of respective resistance genes might also occur via wastewater and terrarium soil. Consequently, possible spreading events of MDR between reptiles and humans seem at the least to be probable since identical clones of extended-spectrum β-lactamase (ESBL)-producing *Enterobacteriaceae* have recently been found in geckos living in close proximity to patients in a hospital in Ghana [12]. Furthermore, *Escherichia coli* isolates from cloacal samples of captive Mexican turtles were found to carry genes for cefotaxime resistance (15.5%), ESBL (5.6%; *bla*_CTX-M-2_ and *bla*_CTX-M-15_), *bla*_CMY-2_ (9.8%), as well as plasmid-mediated quinolone resistance (PMQR) [13]. MDR enterobacteria belonging to ESBL, *bla*_CTX-M_-type β-lactamase, or PMQR genotypes were also detected in other captive as well as wild reptile species [8,14,15,16,17].

Interestingly, a number of studies regards reptiles with respect to potentially bioprospecting due to specific antimicrobial toxins or contents [18,19,20]. On the other hand, reptiles might also cope with facultative pathogenic bacterial loads by a higher activity of antimicrobial molecules that could, for instance, be found in snakes and water monitors inhabiting polluted environments [21].

As outlined elsewhere [22], we initially hypothesized that imprudent antimicrobial use might also have an influence on MDR microbiota in reptiles, especially when they are farmed in or near natural ecosystems. However, this could not be verified in an MDR study on *Acinetobacter* spp. isolated from imported reptiles during border inspections that was recently published by our group [23]. Contrarily, MDR *Acinetobacter* spp. were common in all kinds of import sources. Moreover, even pristine and remote habitats under wildlife protection like the Lesser Antilles and the Galapagos archipelago face contamination with MDR bacteria through anthropogenic activities and the spread of antimicrobial resistance (AMR) genes has become one of the biggest public health threats worldwide [24,25,26].

From the dataset investigated in the current study, preliminary findings on *mcr-1*-positive *Escherichia coli* have previously been published [27]. We here provide results on MDR bacteria of the fraction of *Enterobacterales* that were isolated in a cross-sectional study evaluating 92 shipments of live reptiles, representing 160 batches of single species. Importations and samplings were carried out at the Frankfurt Airport, Germany, which represents the most relevant transshipment point for the trade of live animals worldwide [28].

## 2. Results

### 2.1. Bacterial Isolates and Species Identification by MALDI-TOF MS, TYGS, and JSpec Analysis

A total of 184 samples (163 fecal, 15 skin, three urinate, three mixed fecal/urinate) from 160 sample batches and 92 shipments were obtained. Based on the analysis of MALDI-TOF MS data and after the exclusion of duplicates, 668 *Enterobacterales* isolates were recovered in 98.4% (*n* = 181) of the samples, in 98.9% (*n* = 158) of the sample batches, and in 99.5% (*n* = 91) of the shipments (Table 1). Members of the genus *Klebsiella* were isolated most frequently (*n* = 181; *K. pneumoniae* [*n* = 95], *K. oxytoca* [*n* = 47], *K. aerogenes* [*n* = 39]), followed by the genera *Citrobacter* (*n* = 131; *C. freundii/C. braakii* [*n* = 111], other *Citrobacter* spp. [*n* = 20]), *Escherichia* (all isolates belonged to *E. coli* [*n* = 116]), *Salmonella* (*n* = 69), *Proteus* (*n* = 57; *P. mirabilis* [*n* = 45] and *P. vulgaris* [*n* = 12]), *Enterobacter* (*n* = 52, all *Enterobacter cloacae* complex), *Providencia* (*n* = 20), *Raoultella* (*n* = 11), *Hafnia* (*n* = 9), *Serratia* and *Morganella* (*n* = 8 each), *Yokenella* (*n* = 3), *Leclercia* (*n* = 2), and *Cronobacter* (*n* = 1). From the 668 *Enterobacterales*, the majority were isolated on Gassner agar (*n* = 463) followed by Columbia agar with sheep blood (SBA; *n* = 104), MacConkey agar supplemented with cefotaxime (*n* = 92), and from solid media following growth detection in standard I nutrient broth containing meropenem (*n* = 9). Most of the 668 isolates were recovered from samples from the USA (*n* = 284) and Vietnam (*n* = 135), followed by Uzbekistan (*n* = 42), Ukraine (*n* = 38), Canada (*n* = 33), and 18 other countries (*n* = 136). About two thirds (73.6%) of the isolates were retrieved from *Squamata* (79.4% lizards; 20.6% snakes) and *Testudines* (24.4%; 35.2% turtles; 64.8% tortoises).

### 2.2. Phenotypic Antimicrobial Resistance

#### 2.2.1. Phenotypical Detection of ESBL-/AmpC-β-Lactamase Genes

Among 668 *Enterobacterales* isolates, 99 showed clear growth in a lawn and 30 isolates showed only growth of single colonies on MacConkey agar containing 1 mg/L cefotaxime (Table 1). A combined disk assay test performed on these 129 isolates revealed 41 suspected ESBL producers. These were *E. coli* (*n* = 30), *K. pneumoniae (n* = 7), *Ent. cloacae* complex (*n* = 3), and *C. freundii/C. braakii* (*n* = 1). Thirty-one isolates were identified as putative AmpC producers, which were the following: *E. coli* (*n* = 12), *Citrobacter* spp. (*n* = 9), *Klebsiella* spp. (*n* = 5), and *Ent. cloacae* complex (*n* = 5). Seven isolates did not express an ESBL or AmpC phenotype; these were *Ent. cloacae* complex (*n* = 4), *C. freundii/C. braakii* (*n* = 2), and *K. pneumoniae* (*n* = 1). However, these isolates showed multiple single colonies growing within the zone of inhibition, which implicated their involvement in further analyses. One or more of these 79 isolates in total were found in 27.1% of the samples (*n* = 50), in 28.1% of the sample batches (*n* = 45), and in 40.2% of the shipments (*n* = 50). They were obtained from 15 different countries.

#### 2.2.2. Minimum Inhibitory Concentration Testing

The results from antimicrobial susceptibility testing data of the 79 isolates are depicted in Appendix A. Briefly, since breakpoints for MIC data of reptile *Enterobacterales* isolates are not available, ECOFF values according to EUCAST were used to categorize the bacteria as wild type (WT) or non-wild type (NWT). As had to be expected, all isolates showed high MIC values for penicillin (>2 mg/L), erythromycin (>4 mg/L), tilmicosin (>8 mg/L), and tiamulin (>16 mg/L). Even though no intrinsic resistance to β-lactam antibiotics is known for *E. coli*, all *E. coli* tested showed high MIC values for ampicillin (>8; 100%, *n* = 42), as did the only *Ent. soli* IHIT34087. *Citrobacter* spp., *K. pneumoniae*, and *K. aerogenes* are intrinsically resistant to ampicillin [29]. For amoxicillin–clavulanic acid, 31 of the 54 interpreted isolates were NWT (57%, *n* = 31). *K. aerogenes*, *Ent. cloacae* complex, and various *Citrobacter* spp. are intrinsically resistant to amoxicillin–clavulanic acid and have not been evaluated [29]. For ceftiofur, 72 of 79 isolates were NWT (91.1%, *n* = 72). An NWT phenotpye for the fluorquinolone enrofloxacin could be determined for almost two thirds of all isolates (63.3%, *n* = 50); based on *E. coli* alone, 78% were NWT isolates (*n* = 33 of 42 isolates). In contrast, seven isolates were colistin-NWT (8.9%, *n* = 7), of which five were *Ent. cloacae* and two were *E. coli* (IHIT27704 and IHIT27728, both from Asian grass lizards from Vietnam). The two *E. coli* were described as carrying the *mcr-1.1* gene in a previous study [27]. Almost two thirds of the isolates were identified as NWT for tetracyclines (63.3%, *n* = 50), while for trimethoprim/sulfamethoxazole approximately one half were NWT (50.6%, *n* = 40). For gentamicin, 17.7% of the 79 isolates showed an NWT phenotype (17.7%, *n* = 14). Due to the Micronaut layout used, the interpretation for florfenicol was possible only for *Klebsiella* spp. because the ECOFF for the other species was above the highest concentration measured by the testing system. Consequently, two of 13 *Klebsiella* spp. (15.4%) were NWT.

Of the ten antimicrobial classes tested, β-lactams, cephalosporins, quinolones, polypeptides (colistin), aminoglycosides, tetracyclines, trimethoprim/sulfonamides, phenicoles (only considered for *Klebsiella* spp.), pleuromutilins, and macrolides were most often NWT, regardless of intrinsic resistance against seven of the mentioned classes (27.8%, *n* = 22).

NWT against six drug classes were seen in 20.3% (*n* = 16) of the 79 isolates and NWT against four drug classes were seen in 16.5% (*n* = 13). Eleven isolates each were NWT against five and eight antimicrobial groups (13.9%, *n* = 11), whereas four isolates only showed an NWT phenotype against three of the classes (5.1%, *n* = 4) and two isolates (IHIT27704 and IHIT27728, see above) were NWT against nine of the groups (2.5%, *n* = 2).

### 2.3. Genotypic Antimicrobial Resistance

An ESBL gene was found in nearly half of the 79 *Enterobacterales* genomes examined (49.4%, *n* = 39) (Table 1 and Appendix A). Of these, 29 isolates (74.3%) were identified as *E. coli*, eight (20.5%) as *K. pneumoniae*, and one in each strain (2.6%) as *Citrobacter* sp. and *Ent. cloacae* complex, respectively. While none of the 39 isolates possessed more than one ESBL gene, four isolates co-harbored ESBL and AmpC genes: *E. coli* IHIT27712 obtained from crested forest lizards and green water dragons from Vietnam each carried a *bla*_CTX-M-3_ and *bla*_DHA-1_ gene. Collected from the same sample, *K. pneumoniae* IHIT27711 carried *bla*_SHV-42_ and *bla*_DHA-1_ genes. Strain IHIT34095 (*Citrobacter* sp.) from a leopard tortoise from Ecuador carried *bla*_CTX-M-15_ and *bla*_CMY-46_ genes [30] and IHIT34083 (*Ent. roggenkampii*) from a Horsfield’s tortoise from Uzbekistan possessed a *bla*_CTX-M-15_ and a *bla*_MIR-9_ gene.

#### 2.3.1. ESBL Genes

Overall, the most common ESBL type among the 39 ESBL producers was *bla*_CTX-M_ (89.7%, *n* = 35), while *bla*_SHV_ ESBL genes (*bla*_SHV-12_, *n* = 3; *bla*_SHV-42_, *n* = 1) were only rarely detected.

The *bla*_CTX-M-15_ gene was found predominant (*n* = 16), followed by *bla*_CTX-M-55_ (*n* = 7), *bla*_CTX-M-3_ (*n* = 4), *bla*_CTX-M-27_ (*n* = 3), as well as *bla*_CTX-M-65_ (2.5%, *n* = 2); *bla*_CTX-M-1_, *bla*_CTX-M-130_, and *bla*_CTX-M-14_ were detected in one strain each.

#### 2.3.2. AmpC-β-Lactamase Genes

A total of 42 AmpC-encoding genes were identified in 39 of the 79 *Enterobacterales* genomes (48.1%). These were 12 *Enterobacter* spp. and *Citrobacter* spp. each (31.6% each), 11 *E. coli* each (28.9%), and four *K. pneumoniae* (10.5%).

The most common AmpC family observed among the 42 AmpC cephalosporinases was *bla*_CMY_ (53.7%, *n* = 22). The predominant gene was *bla*_CMY-2_ (29.3%, *n* = 12), whereas *bla*_CMY-3_, *bla*_CMY-46_, and *bla*_CMY-101_ (2.4%, *n* = 1 each) were rarely found. Seven *Citrobacter* spp. carried unknown *bla*_CMY-2_ family class C β-lactamases. Three isolates possessed two different AmpC genes, including IHIT34093 (*E. coli*) from a green water dragon from Vietnam (*bla*_CMY-2_ and *bla*_DHA-1_), IHIT34074 (*Citrobacter* sp.), also from green water dragons from Vietnam, but from another shipment (*bla*_CMY-2_ and *bla*_CMY-46-like_), and IHIT34078 (*Citrobacter* sp.) from a stripeneck musk turtle from the USA (*bla*_DHA-1_ and *bla*_CMY_). Overall, *bla*_DHA-1_ genes were detected in seven of the AmpC-positive isolates, mainly in *Klebsiella* spp. The 12 *Enterobacter* spp. carried chromosomal AmpC-β-lactamase genes, including *bla*_ACT-16_ and *bla*_CMH-like_ (*n* = 4 each), as well as *bla*_ACT-4-like_, *bla*_CMH-9_, *bla*_MIR-3_, and *bla*_MIR-9_ (*n* = 1 each).

A chromosomal mutation in the AmpC promotor gene was detected in *E. coli* IHIT34106. Another *E. coli* (IHIT27704) from a long-tailed lizard from Vietnam obviously lost its plasmid during the course of the investigations, so that the *bla*_CTX-M-55_ and *mcr-1* genes that were present at the beginning were no longer detectable. Four putative AmpC strains, namely IHIT34085 (*E. coli*), IHIT34107 (*Citrobacter sedlakii*), and both *K. aerogenes* (IHIT34075, IHIT44959), could not be verified as AmpC producers based on whole genome sequence analysis. Appendix A provides a detailed overview on ESBL and AmpC genes on isolate basis as well as on the presence of other AMR genes identified among the 79 *Enterobacterales* genomes.

#### 2.3.3. Additional AMR Genes

Nearly all of the isolates (92.4% *n* = 73) carried additional AMR genes. Three or less, four to nine, and ten to 16 additional AMR genes were identified in 25 (31.6%), 31 (39.2%), and 17 (21.5%) of the isolates, respectively. Twenty-seven isolates revealed mutations in the *gyrA*/*parC* region. Single mutations were observed in 16 isolates (*gyrA*: S83T, *n* = 12; S83I, *n* = 1; S83L, *n* = 2; *parC*: S80I, *n* = 1). Seven isolates showed a double mutation (*gyrA* S83L and D87N, *n* = 2; *gyrA* S83I and *parC* S80I, *n* = 5) and 13 isolates, thereof 12 *E. coli*, revealed triple mutations, predominantly a combination of *gyrA* S83L, D87N, and *parC* S80I (*n* = 10). Other genes related with FQR were *aac(6′)-Ib-cr5* (*n* = 9), *qnrB4* and *qnrB6* (*n* = 4 each), *qnrB1* (*n* = 3), *qnrB9* (*n* = 2), *qnrB12* and *qnrB19* (*n* = 1 each), *qnrS1* (*n* = 20), *qnrS13* (*n* = 1), and one novel *qnrS* gene, that was identified in an AmpC-producing *E. coli* from a Chinese water dragon from Vietnam. Of note, we determined a quinolone resistance gene *qnrVC4* in one *Citrobacter* sp. isolated from an Asian grass lizard from Vietnam. The gene was located on a 165 kb plasmid with 99.9% similarity to the unnamed plasmid (GenBank: CP038466.1) of *Aeromonas hydrophila* strain 23-C-23, which was isolated from an outbreak of fatal diarrhea among farm-raised vipers in China [31]. Notably, resistance determinants targeting against quinolones are listed among critical important antimicrobials for human medicine by the WHO [32]; target-mediated resistance is the most common and clinically significant form of resistance, which is caused by specific mutations in gyrase and topoisomerase IV that weaken interactions between quinolones and these enzymes. Overall, 69 (87.3%) isolates carried either a gene linked with FQR or revealed a mutation in the *gyrA*/*parC* region, which leads to a reduced minimal inhibitory concentration to quinolones.

Antimicrobial resistance genes against tetracyclines (64.5% of the 79 isolates positive), aminoglycosides (57%), trimethoprim (50.6%), other β-lactam antibiotics (45.6%), and sulfonamides (41.8%) were detected frequently. For tetracylines, *tet(A)* (*n* = 43) was predominant, followed by *tet(B)* (*n* = 7), *tet(D)* (*n* = 3), *tet(C)*, and *tet(M)* (*n* = 1 each). Genes conferring resistance to different aminoglycoside antibiotics were the following: *aph(6)-Id* (32.9%), *aph(3′)-Ib* (31.6%), *aadA2* (*n* = 13.9%), *aadA1* (*n* = 12.7%), *aac(6′)-Ib-cr5* (*n* = 11.4%), *aph(3′)-Ia* (8.9%), *aac(3)-IIe* (7.6%), *aac(3)-IId* (6.3%), *aadA5* (5.1%), *aac(6′)-If* and *aac(6′)-Ib* (2.5% each), *aac(3)-IVa*, *aadA16*, and *aadA22* (1.3% each). Trimethoprim genes identified were *sul1* (17.7%), *sul2* (35.4%), and *sul3* (10.1%).

Genes conferring resistance to phenicoles (24.0%), including *catA1* (1.3%), *catA2* (6.3%), *catB3* (2.5%), *cmlA1* (6.3%), and *cmlA5* (2.5%) and to macrolides (15.2%, *n* = 12), including *mph(A)* (15.2%) and *erm(B)* (1.5%), were detected in somewhat smaller numbers. Only two isolates carried lincosamide resistance gene *Inu(F)* (2.5%). The *mcr-1* gene, which plays a role in plasmid-mediated colistin resistance, was found in *E. coli* IHIT27728 from a long-tailed lizard from Vietnam (1.3%; [27]). Genes conferring resistance to rifampicin, which is often used to treat human tuberculosis, were detected in six isolates (7.8%; *n* = 2 *arr-2*; *n* = 4 *arr-3*). Carbapenemase genes were not detected.

### 2.4. Plasmid Characterization

In more than half (55.7%) of the isolates examined, the ESBL/AmpC genes could be localized on plasmids. Apart from 13 cases, where the Inc type could not be determined (*n* = 9 ESBL-*E. coli*, *n* = 1 AmpC-*E. coli*, *n* = 1 ESBL-*K. pneumoniae*, and *n* = 2 AmpC-*K. pneumoniae*), in silico prediction revealed IncIγ/K1 (*n* = 9) as the most common ESBL/AmpC plasmid type. It was found in *E. coli* from the Americas (*n* = 6), Asia (*n* = 2), and South Africa (*n* = 1) collected from different animal species and carrying various ESBL/AmpC genes (six × *bla*_CMY-2_ and one each of *bla*_CTX-M-1_, *bla*_CTX-M-65_, and *bla*_SHV-12_). IncFIB plasmid types, either alone or in combination with other Inc types, were also common (*n* = 9) and were predicted in *K. pneumoniae* (*n* = 4) and *E. coli* (*n* = 3) mainly from Asia (*n* = 6) or South America (*n* = 3). Additional plasmid types observed were IncC (ST48, *E. coli, bla*_DHA-1_, Vietnam; ST398, *E. coli*, *bla*_CTX-M-15_, Uzbekistan; ST401, *Citrobacter*, *bla*_CMY-2_, Vietnam; and ST401, *Citrobacter, bla*_CMY-2_, Vietnam), IncX1 (ST101, *K. pneumoniae, bla*_DHA-1_, Vietnam and ST1087, *K. pneumoniae*, *bla*_DHA-1_, Vietnam), IncX2 (ST1408, *E. coli*, *bla*_CTX-M-55_, Vietnam and ST2705, *E. coli*, -*bla*_CTX-M-55_, Vietnam), IncP (ST757, *E. coli*, *bla*_CTX-M-130_, Vietnam), IncR (ST18, *Citrobacter*, *bla*_DHA-1_, USA), IncHIB (ST15687, *E. coli*, *bla*_CTX-M-27_, Nicaragua), and IncHI2A (ST1011, *E. coli*, *bla*_CTX-M-55_, Vietnam). Overall, plasmid types were very diverse with respect to ESBL/AmpC type, bacterial species, sequence types, hosts, and countries of origin (Figure 1, Figure 2 and Figure 3).

### 2.5. MLST

From the 79 isolates that were selected for WGS, 40 isolates could be assigned to a previously existing ST, whereas the remaining isolates were either novel STs (*n* = 14) or were not typeable, e.g., due to the lack of one of the seven alleles used for MLS typing in the respective genome (Figure 1, Figure 2 and Figure 3; Appendix A). The STs found were predominantly singleton findings. However, isolates with certain STs were found more frequently, such as ST48 (*n* = 4; *E. coli*), ST182 (together with ST182-like; *n* = 3; *Ent. hormaechei* ssp. *xiangfangensis*), as well as ST432 (*Ent. cloacae* ssp. *cloacae*), ST58 (*E. coli*), ST167 (*E. coli*), ST226 (*E. coli*), and ST401 (*C. freundii/C. braakii*) that were found in duplicate (each *n* = 2). Interestingly, more frequently encountered STs did not necessarily match with respect to host species, country of origin, or production system. But, two out of three *bla*_ACT-16_-positive *Enterobacter* isolates representing closely related lineages among ST182 were isolated from shipments containing members of the genus *Physignatus* that were wild-caught specimens from Vietnam, and this was also true for two out of four *E. coli* of ST48. Both ST401 *C. portucalensis/C. freundii* were AmpC producers and were also found in wild-caught specimens from Vietnam (two shipments with *Physignatus cocincinus*/*Leiolepis belliana* and *Takydromus sexlineatus*, respectively). Eventually, both *Ent. cloacae* representing ST432 were detected in captive-bred leopard geckos (*Eublepharis macularius*) in the same sample from China.

In contrast, there were various batches (*n* = 6) in which several isolates of the same bacterial species but with different STs and often different ESBL/AmpC genes were found. Five of these combinations were found in *E. coli*: In batch 11 (wild-caught agamas, Vietnam), we identified isolates of four different STs (ST131, ST226, ST1408, and ST3014) and various ESBL/AmpC types (*bla*_CTX-M-27_, _-55_, _-65_, and *bla*_CMY-2_) Three isolates each, differing by sequence types and ESBL/AmpC types, were also found in batch 70 (farm-bred Horsfield’s tortoises, Uzbekistan), 92 (captive-bred /farm-bred Parson’s chameleons, USA), and 160 (wild-caught painted wood turtles, Nicaragua). Also, for *K. pneumoniae*, we detected different isolates (ST16 and ST307), both carrying a *bla*_CTX-M-15_ gene, in batch 129 (farm-bred leopard tortoises, Ecuador).

A comparison based on core genome MLST alleles for *E. coli* (Figure 1) and *K. pneumoniae* (Figure 2) and based on concatenated alleles of the seven-gene MLST scheme for *Enterobacter* sp. (Figure 3) underlined the overall high diversity of ESBL/AmpC isolates within the different species. However, we identified *E. coli* and *Enterobacter* spp. from different countries and/or animal hosts clustering together, as depicted in the following figures.

## 3. Discussion

The aim of this study was to assess the diversity and antimicrobial resistance properties of enterobacteria in this vertebrate class during importation for pet trade or animal experiments. In three previous publications, we have focused on two colistin-resistant *E. coli*, *bla*_CTX-M-15_-producing *K. pneumoniae* ST307, and on *Acinetobacter* spp. from the same importations [23,27,30]. We hypothesized that poor hygienic and environmental conditions in breeding facilities might be correlated with increased rates of MDR bacteria and their global distribution in imported reptiles [23]. However, our previous results suggested a rather poor correlation in this regard, because MDR bacteria from the genus *Acinetobacter* were regularly found also in samples from wild-caught specimens, thereby merely indicating environmental pollution or a spillover of AMR directly before export [23]. With respect to the significance of unraveled AMR, it is important to emphasize that reptiles have been sampled directly at the point of entry during border inspection in order to prevent bias of the detected microbiota from inland sources. The Frankfurt Airport in Germany is regarded as one of the most important hubs in animal importation worldwide. In 2022, 438 shipments with 662,460 animals declared as ‘reptiles’ were registered for import into the European Union (EU), from which 277 shipments (394,145 animals) remained in the EU, whereas 73 shipments were sent in transit with a total number of 32,465 live reptiles [33].

Principally, the importation of wild-caught animals has been widely banned. Nevertheless, as indicated in Appendix A, a number of the imported species under study were declared as specimens that were collected in the wild. Across all 668 isolates, a considerable portion of isolates (*n* = 257) were regarded as wild-caught specimens based on the expertise of three independent reptile experts of the German Herpetological Society (DGHT). Conversely, another third of isolates (*n* = 255) originated from offspring in captivity and a slightly smaller amount were considered as farm-bred reptiles (*n* = 151). For five isolates of *Diploglossus monotropis*, no assessment was possible. However, flowing transitions were found between these three groups and an unequivocal differentiation was not possible in all cases. Furthermore, it was not possible to differentiate between “farming” (commercial trade of offspring from parental generation from the wild by permanent in situ or ex situ husbandry) and “ranching” (commercial trade of in situ-produced offspring following collection of gravid females from the wild) practices in the latter group due to general paucity of information. The same was true with some of the metadata on the source of origin, e.g., when African leopard tortoises were declared as being farm-bred in Ecuador, where no such farms are known to exist.

In the present study, 92 shipments with 160 sample batches containing live reptiles were tested for the presence of MDR enterobacteria. Four genera of the *Enterobacteriaceae* were finally chosen out of the total amount of *Enterobacterales* for in-depth analysis, because they represented the four most commonly encountered genera that are very well known for their role as potential MDR bacteria as well as for their highly diverse array of β-lactamases [34]. The vast majority of isolates collected from MacConkey agar supplemented with 1 mg/L cefotaxime displayed ESBL- or AmpC-encoding genes (Table 1). The presence of these genes could not be allocated to a specific geographical region and their occurrence was noted among all investigated reptile classes and production lines. Furthermore, ESBL producers were highly diverse with the finding of different *bla*_CTX-M_ enzymes within isolates of the same bacterial species and in the same shipment of a single reptile species. This was exemplarily seen for *E. coli* IHIT26863 (*bla*_CTX-M-55_, ST4539) and IHIT27723 (*bla*_CTX-M-14_, ST746) that were obtained from two captive-bred Parson’s chameleons imported from the USA. On the other hand, unexpected doublings were noted in, e.g., two *K. pneumoniae* from the same shipment of wild-caught *Calotes emma/Physignatus cocincinus* lizards from Vietnam that both comprised a *bla*_DHA-1_ gene (IHIT27711 and IHIT44958), but on different plasmids and in a different phylogenetic background (ST1087 and ST709). In the latter case, a spillover of AMR genes during collection or at intermediate dealers cannot be ruled out. We identified equal numbers of ESBL and AmpC cephalosporinase-positive isolates (*n* = 39 each). The set of ESBL types was dominated by *bla*_CTX-M_ (45.6%; predominantly *bla*_CTX-M-15,_ and *bla*_CTX-M-55_), followed by *bla*_SHV_ (14.6%; predominantly *bla*_SHV-12_). AmpC-encoding genes were detected in 49.4% of the 79 isolates and here, *bla*_CMY_ types (48.7%; predominantly *bla*_CMY-2_) dominated, followed by *bla*_DHA-1_ (17.9%), *bla*_CMH_ (12.8%), and *bla*_ACT_ (12.8%), respectively. Interestingly, other studies predominantly found *bla*_CMY-1_ and *bla*_CMY-2_ in Spanish pond turtle feces, but *bla*_CTX-M-15_ was found only in small numbers in non-*Enterobacteriaceae* [16]. Captive-reared Mexican pond turtles harbored fewer amounts of ESBL- (*bla*_CTX-M-2_ and *bla*_CTX-M-15_)/AmpC (*bla*_CMY-2_)-producing *E. coli* (5.6% and 9.8%, respectively) compared to our data for *E. coli* (25.0% and 8.6%, respectively) [13].

Although there are various reports on the occurrence of *Salmonella* spp. carrying *bla*_CTX-M_ genes, e.g., from Southeast Asia from poultry and pork or from hospitalized horses from Israel [35], ESBL- or AmpC-producing *Salmonella* spp. were not detected in this study, which is in line with another study on *Salmonella* spp. from reptiles [11]. Of note, all ESBL- and AmpC-carrying isolates detected in this study were isolated only from selective media, namely MacConkey agar supplemented with cefotaxime. Thus, these isolates might have been overlooked during routine diagnostic procedures. This is a known phenomenon also for fecal samples from other animals and underlines the necessity to include antibiotic selective media to truly identify the antimicrobial resistance burden in reptiles [36].

Among the typeable ESBL/AmpC plasmids, IncIγ/K1 was the most common predicted type among the 79 isolates examined, and this has rarely been described to date ( Table 2). IncIγ/K1 plasmids belong to the IncI complex that can be divided into the seven subgroups IncI1, IncI2, IncIγ, IncB/O, IncK1, IncK2, and IncZ. In previous studies, IncIγ/K1 plasmids were found in *bla*_CTX-M_-positive *E. coli* and *K. pneumoniae* from human patients in China [37]. There is a report from Peru linking IncIγ/K1 plasmids to commercial guinea pig production and *Salmonella* Typhimurium infections [38]. To our knowledge, this is the first description of an IncIγ/K1 plasmid from an isolate from South Africa. IncFIB plasmids have been repeatedly described as carriers of *bla*_CTX-M_ genes, e.g., in *E. coli* with *bla*_CTX-M-15_ from pediatric patients from Mexico [39], *E. coli* with *bla*_CTX-M-55_ from two hospitals in China [40], or *K. pneumoniae* also with *bla*_CTX-M-15_ from a hospital in Egypt [41]. IncFIB-carrying *E. coli* with chromosomally encoded *bla*_CTX-M-65_ genes were found in samples of beef and pork from Portugal [42]. In 2016, *E. coli* carrying IncFIB and *bla*_CTX-M-15_ genes were detected in turtles from Mexico [13]. Likewise, from China comes a report on *E. coli* with *bla*_CTX-M-55_ on an IncFIB plasmid from diseased captive giant pandas [43] or *K. pneumoniae* with *bla*_CTX-M-14_ from a red kangaroo [44]. In the latter study, a *K. pneumoniae* was also found with *bla*_CTX-M-3_ on an IncFII(K) plasmid.

In this study, there were several clusters, in which IncFIIB plasmids or combinations thereof were isolated from reptiles from the same country of origin and partly from the same species and on the same day, but which came from different bacterial isolates with different STs. This fosters the assumption that individual plasmids with resistance genes might be spread in different *Enterobacterales* species in the same reptile farms. It is also tempting to speculate that environmental pollution with AMR genes might have caused the acquisition of MDR bacteria in wild-caught animals.

Furthermore, a number of reports from various bacterial species on IncC plasmids, which carry *bla*_CMY-2_ genes, have their origin in Asia and were primarily reported from livestock animals. For example, several studies revealed IncC plasmids in *E. coli*, *K. pneumoniae*, *Salmonella enterica*, *Proteus mirabilis*, and *C. braakii* from chickens and pigs from China [45], *E. coli* from food-producing animals from southern China [46], or *E. coli* from healthy broilers from Japan [47]. There is also a report on *C. freundii* and IncC types with *bla*_CMY_ [48]. *Bla*_CTX-M-15_ on an IncC plasmid was detected in *E. coli*, in which they appear to be less established in combination with *bla*_CTX-M_ than in other plasmid types [49]. There is always debate as to whether the off-label use of cephalosporins is relevant as an influencing factor on the development of resistance in poultry flocks, for example in reports from Japan [50], and the question arises here whether a contaminated environment could possibly lead to introductions into wild reptile species.

From Africa, there have been a number of reports of *E. coli* from camels as well as from monkeys with IncY plasmids carrying *bla*_CTX-M-15_ [51,52]. IncX2 plasmids have not yet been described as a carrier of *bla*_CTX-M_ genes. An *E. coli* isolate with an IncX1 plasmid without resistance genes and additional plasmids carrying *bla*_CTX-M-15_ and *bla*_DHA-1_ was detected in a litter of puppies that died from this infection in Italy [53]. In a Malaysian pangolin (*Manis javanica*), which is often hunted illegally, an IncX1 plasmid in an *E. coli* isolate was associated with antibiotic resistance [54]. In this study, plasmids with the same Inc types from the same country of origin, but with identical or different shipment numbers or samples, were repeatedly identified. This could again be indicative of a spread of resistance plasmids between different bacterial isolates within a reptile’s holding facility or environment, especially when only a few or even the same dealer export from a certain country.

**Table 2 antibiotics-13-01230-t002:** ESBL-/AmpC-β-lactamase genes and plasmid types found in this study and their finding in reptiles according to the literature.

**ESBL/AmpC Genes Determined in This Study**	**Previous Studies Reporting the Study of ESBL/AmpC Genes in Reptiles (Among Others)**
*bla* _CTX-M-like_	[16]
*bla* _SHV-12, -42_	[13]
*bla* _CMY-like_	[30]
*bla* _DHA-1_	[55]
*bla* _CMH-like_	[56]
*bla* _ACT-like_	[57]
*bla* _MIR-like_	[58]
*bla* _CFE-1_	[59,60,61,62]
**Plasmid Type**	**Previous Studies Reporting the Study of Plasmid Types in Reptiles (Among Others)**
IncIγ/K1	[37,38,63]
IncFIB	[13,39,40,41,42,43,44]
IncC	[45,46,47,48,49]
IncY	[51,52,64]
IncX2	[65,66,67]
IncX1	[53,54]

With respect to the diversity of sampled import countries, host species, and time periods over one year, we expected a rather small level of relatedness between isolates in our samples, but we anticipated matches regarding ST to dominant bacterial lineages that have been described in humans or other animal classes on a local or global scale. Indeed, among *E. coli*, ST48, ST58, ST167, and ST226 were found in at least two isolates, while ST182 and ST401 appeared two times among *Enterobacter* sp. and *Citrobacter* sp. isolates, respectively. *Ec*-ST48 could be found in imports from Vietnam, the USA, and Colombia, and in Vietnam, this ST was among those strains isolated from flies, livestock, and humans that additionally carried *mcr-1* genes [68]. Some ST58 isolates are known to belong to high-risk *E. coli* clones with high virulence potential. In the USA, where one of the isolates from this study was found, this ST was previously isolated in swine production, retail meat products, and as foodborne zoonotic *E. coli* [69,70]. A study from Mexico has detected ST58 in *bla*_CTX-M-15_-positive *E. coli* [13], whereas the two isolates from our study were AmpC producers (*bla*_CMY-2_-positive). ST167 isolates were among the globally distributed carbapenemase-producing *E. coli,* and Egypt as well as India, France, and the United States were identified as crucial transmission hubs for this ST [71]. However, isolate IHIT27727 from a red-headed rock agama from Egypt lacked genes for carbapenem resistance. *E. coli* ST226, albeit also not yet found in reptiles, is known to occur in bovines, chickens, pigs, wild boars, Barbary macaques, beef, and humans and has been reported from Portugal, Africa (Algeria, Egypt, Nigeria), Thailand, and China [59,72,73]. Interestingly, three out of four isolates of *Ent. hormaechei* ssp. *xiangfangensis* belonging to ST182 were found in wild-caught agamid lizards of the genera *Calotes* and *Physignathus*. This ST has been reported from canine and feline opportunistic infections in Thailand [74], from hospital outbreaks in Czechia, Greece, Iran, Kenya, Mexico (*bla*_NDM-1_-carrying; [75,76,77,78]), and Argentina (*bla*_VIM_-carrying; [79]), respectively, in *bla*_IMP_-resistant *Enterobacter* circulating in Spain [80], but not yet from reptilian hosts. *Citrobacter* ST401 have been rarely documented with only one isolate from the next closely related vertebrate class amphibians [81]. Despite lacking the *mcr-9* gene that was present in the Chinese bullfrog strain, isolates from this study from three different wild-caught lizard shipments from Vietnam (IHIT34074, IHIT34079) contained various AMR genes, including *bla*_CMY-2_, *bla*_CMY-46-like_*, bla*_OXA-10_, *cmlA5*, *floR*, *mph(A)*, *aac(6′)-Ib*, *aph(3′)-Ia*, *qnrB6*, *qnrVC4*, *aph(3″)-Ib, aph(6)-Id*, *aadA1*, *sul2*, *tet(A)*, and *dfrA14*, most of them co-localized on an IncC plasmid. Of note, the quinolone resistance gene *qnrVC4*, which was originally identified in *Aeromonas* spp., has rarely been observed in *Enterobacterales* so far. The first report in a *C. freundii* isolate, namely from human urine in China, dates back to 2019. Here, *qnrVC4* was co-located with *bla*_KPC-2_ on a hybrid plasmid of 35 kb in size [82]. However, the two *C. freundii* from our study carry the integron-mediated *qnrVC4* gene on a 165 kb IncC plasmid with 99.7% similarity to the plasmid from original *Aeromonas hydrophila* strain WCX23 [31].

*E. coli* ST131 with, among others, *bla*_CTX-M-27_ and *K. pneumoniae* ST307 carrying *bla*_CTX-M-15_, are often described as high-risk clones worldwide [56]. In silver gulls from Australia, in which the above-mentioned clones *E. coli* ST131 and *K. pneumoniae* ST307 were found, a connection was found between the occurrence of critically important antimicrobial-resistant *E. coli* and *K. pneumoniae* and the proximity to human settlements in contrast to the occurrence in remote places [57]. A turtle has earlier been reported as a carrier of *K. pneumoniae* ST307 for the first time [58] as has been an *E. coli* ST131 among 24 *E. coli* strains from various reptiles [83]. The isolates IHIT340297 *K. pneumoniae* ST307 with *bla*_CTX-M-15_ from farm-bred panther turtles from Ecuador and IHIT27287 *E. coli* ST131 with *bla*_CTX-M-27_ from wild-caught forest garden lizards/green water dragons from Vietnam described in the present study could therefore represent another mosaic piece in the global spread of these clones or even be discussed as sentinels for a contaminated environment.

When assessing the zoonotic potential, the following considerations should be taken into account. In Europe, reptiles are primarily kept as pets, so it can be assumed that the imported reptiles in this study are also intended for this purpose and are not intended for human consumption as in some of their countries of origin. *Salmonella* in particular is common in reptiles and can be transmitted to humans due to inadequate hygiene standards [84]. It would also be possible for MDR bacteria to be transmitted to humans when keeping reptiles in terrariums, for example during handling or cleaning work. In general, studies are increasing that deal with the occurrence of MDR bacteria in reptiles, for example in young sea turtles in Mexico [85] or clinical samples from various reptile species on the Iberian Peninsula [86]. In addition to a possible spillover of MDR bacteria, a transfer of so-called high-risk clones detected in this study, such as *E. coli* ST131 and *K. pneumoniae* ST307, to humans may also be possible. The global trade in reptiles, similar to other globally traded animals or foods, migratory birds, or air travelers themselves, creates another possible vector that can lead to rapid and global spread of certain MDR bacteria or certain clones.

## 4. Materials and Methods

### 4.1. Sample Collection

The samples were collected as previously described [23]. Briefly, samples were drawn from imported reptiles at Frankfurt Airport, Germany between July 2013 and May 2014. The sampling was carried out in cooperation with authorized veterinarians from the Border Control Post during import control at the Frankfurt Animal Lounge directly after arrival of the animals. According to the guidelines of the International Air Transport Association (IATA), the different shipments contained either one or several species that were transported in separate boxes. The investigated reptiles were assigned to the orders *Squamata* (lizards and snakes) and *Testudines* (turtles and tortoises). A total of 92 shipments were included, and each reptile species per shipment was considered as one sample batch (*n* = 160), from which one or more samples were taken (*n* = 184). For analysis, feces (90% of the specimens; *n* = 163), remnants of ecdysis (shed skin), and urinate (separate or mixed) were retrieved. The shipments originated from 23 different countries in North America (USA and Canada), Central and South America (Brazil, Ecuador, El Salvador, Guatemala, Guyana, Colombia, Nicaragua, and Panama), Asia (China, Japan, Uzbekistan, Turkey, and Vietnam), Africa (Egypt, Madagascar, Mozambique, Uganda, South Africa, and Tanzania), and Europe (Ukraine and Macedonia). High numbers of shipments came from the USA (38.0%), Vietnam (10.9%), Uzbekistan (9.8%), and Ukraine and Canada (5.4% each). As already described in a previous publication, the reptiles investigated in this study were assigned to different natural sources, including captive-bred (33.8%), wild-caught (28.3%), and farm-bred species (19.8%), whereas a number of species could not be clearly assigned [23].

### 4.2. Bacterial Isolates and DNA Extraction

After sampling, specimens were stored in sterile sample tubes at 4–7 °C. Transport to the laboratory was carried out no later than 24 h after sampling. Fecal and urinate samples were directly inoculated on blood agar (Merck, Darmstadt, Germany, supplemented with 5% sheep blood; SBA), water-blue metachrome-yellow lactose agar (Gassner; Oxoid, Wesel, Germany), and MacConkey agar (Oxoid) supplemented with 1 mg/L cefotaxime (Sigma-Aldrich/Merck, Darmstadt, Germany). Resuspension of shedding remnants was performed in 0.9% NaCl before inoculating the same media with 100 µL aliquots. All media were cultured at 37 °C for 24 h. Each sample was also cultivated in 3 mL standard I nutrient broth (Roth GmbH + Co. KG, Karlsruhe, Germany) containing a 10 µg meropenem disc (Mast Group Ltd., Reinfeld, Germany). In case of visible growth after 24 h of incubation at 37 °C, 50 µL were streaked onto SBA and Gassner agar. In addition, all specimens were cultivated in 5 mL nutrient broth supplemented with bovine serum (Merck KGgA, Darmstadt, Germany) for 24 h at 37 °C for concurrent enrichment. Bacterial colonies with different morphologies were subcultured for species identification. All isolates were conserved at −80 °C in a liquid nutrient broth supplemented with 20% glycerol (Merck KGgA) for further investigations. DNA extraction was performed with a MasterPure^TM^ DNA Purification Kit according to the manufacturer’s instructions (Biozym Scientific GmbH, Hessisch Oldendorf, Germany).

### 4.3. Bacterial Species Identification

All isolates grown on non-selective and selective media that were suspected to be Gram-negative bacteria were analyzed by MALDI-TOF MS using a microflex LT mass spectrometer (Biotyper version 3.3.1.0 with MBT compass software, version 3.0). Bacterial material was transferred to the steel target using the direct transfer protocol according to the manufacturer’s instructions (MALDI Biotyper; Bruker Daltonik, Bremen, Germany). Correct species-level identifications were supposed when the first and second matches showed the same species with log scores of >2.0, or when the best match showed a bacterial species with a log score of >2.0 and the second match a different species with a log score of <2.0. Each isolate was tested in duplicate to verify the results. If morphologically different bacterial phenotypes of one bacterial species assigned to the *Enterobacterales* order were contained, these were stored so that different isolates of this species originate from the same sample. All isolates were stored at −80 °C until further investigations. Exceptions were made for isolates of the *Enterobacter* (*Ent.*) *cloacae* complex (*Ent. cloacae*, *Ent. asburiae*, *Ent. hormaechei*, *Ent. kobei*, *Ent. ludwigii*) as well as for *Citrobacter* (*C*.) *freundii* and *C. braakii*, because respective species give rise to incorrect species-level identifications by MALDI-TOF MS [87]. In these cases, only one isolate per batch was conserved.

Species identification of putative ESBL- and AmpC-β-lactamase-producing bacteria (see Section 2.4) was further performed based on whole genome sequence analysis (see Section 2.5). For this, draft genome sequence data were uploaded to the TYGS server, a free bioinformatics platform available at https://tygs.dsmz.de (accessed on 16 October 2024). The taxonomic status of *Enterobacter* and *Citrobacter* spp. was additionally assessed at the whole genome level by performing average nucleotide identity (ANI) calculations based on BLAST+ using ANIb implemented in JSpeciesWS (© 2014–2023 Ribocon GmbH, Bremen, Germany—version: 4.0.2).

### 4.4. Antimicrobial Susceptibility Testing and Identification of Extended-Spectrum-β-Lactamase (ESBL)- and AmpC-β-Lactamase-Producing Bacteria

All *Enterobacterales* isolated from non-selective media and assigned to the genera *Klebsiella* (*K.*), *Enterobacter*, *Citrobacter*, and *Salmonella* (*S*.) as well as to the species *E. coli* were streaked onto MacConkey agar (Oxoid) containing 1 mg/L cefotaxime (Sigma-Aldrich/Merck). In case of growth, either after direct cultivation of the sample on selective medium or after re-streaking isolates from the non-selective approach, phenotypic confirmatory combined disc testing according to CLSI guidelines was performed using antimicrobial discs containing cefotaxime (30 µg), ceftazidime (30 µg), cefotaxime/clavulanic acid (30 µg/10 µg), and ceftazidime/clavulanic acid (30 µg/10 µg) [29].

For all isolates displaying growth on MacConkey agar supplemented with cefotaxime (1.0 mg/L) and showing a conspicuous resistance pattern in the confirmatory test, minimal inhibitory concentration (MIC) values were determined. We used broth microdilution susceptibility testing with a commercial test kit layout for livestock (Micronaut/Bruker according to the guidelines of the working group antimicrobial resistance of the German Veterinary Society [DVG]). This layout contained 16 different antimicrobials (in µg/mL: amoxicillin/clavulanic acid [2/1–16/8], ampicillin [0.125–16], ceftiofur [1–4], colistin [0.5–2], enrofloxacin [0.063–1], erythromycin [2–4], florfenicol [1–8], gentamicin [2–8], penicillin G [0.063–2], tetracycline [1–8], tiamulin [8–16], tilmicosin [4–16], and trimethoprim/ sulfamethoxazole [0.25–4.75/2–38]).

### 4.5. Whole Genome Sequencing

Isolates belonging to the species *E. coli* and to the genera *Citrobacter*, *Enterobacter*, and *Klebsiella* and provisionally determined as putative ESBL- or AmpC-β-lactamase producers were whole-genome-sequenced (*n* = 79). We used the Nextera XT Library Preparation Kit (Illumina GmbH, Munich, Germany) to prepare the sequence libraries. Sequencing was performed on an Illumina MiSeq platform (Illumina, Inc., San Diego, CA, USA) using paired-end sequencing. After quality control using default parameters, adapter-trimmed reads were assembled using SPAdes v3.13.1. Draft genomes were annotated using Prokka v1.13.

### 4.6. Location of ESBL- and AmpC-β-Lactamase Genes Using S1-PFGE and WGS

The genomic location of ESBL- and AmpC-β-lactamase genes was verified by S1 nuclease (Fermentas GmbH, St. Leon-Rot, Germany) digestion followed by electrophoretic separation and Southern blot hybridization. A digoxigenin-labeled DNA probe (DIG luminescent detection kit, Boehringer Mannheim GmbH, Mannheim, Germany) was utilized to target PCR fragments specific for *bla*_CMY_, *bla*_CTX-M_, *bla*_SHV_, *bla*_DHA_, and *bla*_CMY_ genes by using primers provided in Appendix A [88,89,90]. We initially tested 34 *E. coli*, 10 *K. pneumoniae*, 2 *C. freundii/C. braakii*, and 2 *Ent. cloacae* complex isolates in this approach, which were positive in previous PCR for ESBL/AmpC genes (*n* = 48 in total). After the localization of the ESBL/AmpC gene was investigated in 48 isolates using S1 nuclease and PFGE, all 79 isolates were whole-genome-sequenced in a later phase of the study. Using WGS data, the localization of the ESBL/AmpC gene could also be analyzed in the remaining 31 isolates. In detail, an in silico search of whole genome sequences was performed with the bioinformatics tool “Chromosome and plasmid overview” implemented in Ridom Seqsphere+ (version 10.0.2 [2024-06]; https://www.ridom.de/seqsphere; accessed on 16 October 2024).

### 4.7. Determination of AMR Genes, Chromosomal Mutations, and Plasmid Types

Additional AMR genes besides ESBL/AmpC genes were identified from the genomic data by using the online platform tool ResFinder version 4.1. Additionally, and as a cross check, we utilized the Resistance Gene Identifier (RGI) software (version 6.0.3) from the Comprehensive Antimicrobial Resistance Database (CARD, version 3.3.0) to unravel resistance genes. Furthermore, NCBI AMR Finder software (version 4.0.3) implemented in Ridom SeqSphere was used to identify AMR genes and to identify their putative locations on plasmids or chromosomes (see Figure 1, Figure 2 and Figure 3).

### 4.8. Multilocus Sequence Typing (MLST) of Enterobacterales Isolates and Phylogenetic Grouping of E. coli

Multilocus sequence type (ST) analysis was performed for all whole-genome-sequenced *E. coli*, *Klebsiella* spp., *Ent. cloacae* complex, and *Citrobacter* spp. For the species *Ent. cloacae*, *K. aerogenes*, and *C. freundii*/*C. braakii*, the MLST schemes provided by PubMLST were used (https://pubmlst.org/bigsdb?db=pubmlst_ecloacae_seqdef; https://pubmlst.org/bigsdb?db=pubmlst_kaerogenes_seqdef; https://pubmlst.org/bigsdb?db=pubmlst_cfreundii_seqdef, accessed on 16 October 2024). 

Isolates of the species *K. pneumoniae* were subjected to an MLST scheme provided by Institute Pasteur, France (bigsdb.pasteur.fr/klebsiella), and for *E. coli* the scheme on enterobase.warwick.ac.uk/species/index/ecoli was employed. Novel alleles and STs found were submitted to the appropriate database curator to include new alleles and STs in the database.

Genome sequence data of *E. coli* were further analyzed in silico to classify isolates into one of the eight phylogenetic groups (A, B1, B2, C, D, E, F, and G) or into a cryptic clade using the refined ClermonTyping method, based on the in vitro PCR assay targeting *chuA*, *yjaA*, *TspE4.C2*, *arpA*, and *trpA* (http://clermontyping.iame-research.center/, accessed on 25 March 2024).

## 5. Conclusions

This study shows that imported reptiles are often carriers of AMR bacteria; one or more of the 79 isolates examined in this study were found in >27% of the samples or in >40% of the shipments. Many ESBL and AmpC genes were detected, of which *bla*_CTX-M_ and *bla*_CMY_ were the most common. Among them were also high-risk clones relevant to human medicine: ST131 *E. coli bla*_CTX-M-27_ and ST307 *K. pneumoniae bla*_CTX-M-15_. Many isolates (>90%) carried additional AMR genes against other classes of antibiotics. Against fluoroquinolones, for example, either chromosomal or plasmid-mediated resistance factors were found in 87% of the isolates. As expected from such a broad sample source, many different known and new STs were found, but also some identical STs from different samples and with human medical relevance. Since the imported animals are primarily intended for the pet market, possible zoonotic transmissions between reptiles and their keepers are conceivable without proper hygiene measures. The imported reptiles also represent a possible additional vector for the global spread of AMR.

## Figures and Tables

**Figure 1 antibiotics-13-01230-f001:**
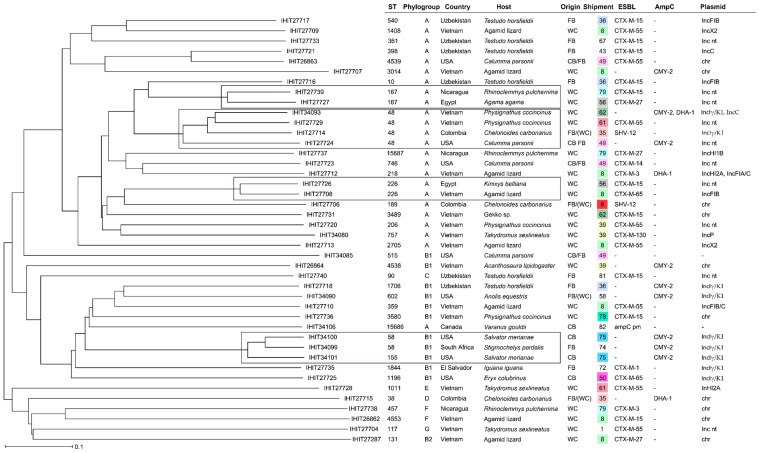
NJ tree of 42 *E. coli* based on pairwise distance calculation of 2513 cgMLST alleles. chr = chromosomal; nt = not typeable; pm = promoter mutation; CB = captive breed; FB = farm breed; WC = wild-caught.

**Figure 2 antibiotics-13-01230-f002:**
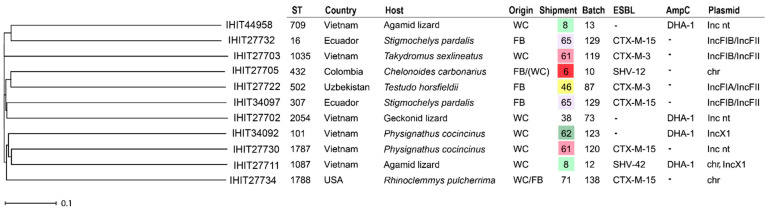
NJ tree for 11 *K. pneumoniae* based on pairwise distance calculation of 2358 cgMLST alleles. chr = chromosomal; nt = not typeable; pm = promoter mutation; CB = captive breed; FB = farm breed; WC = wild-caught.

**Figure 3 antibiotics-13-01230-f003:**
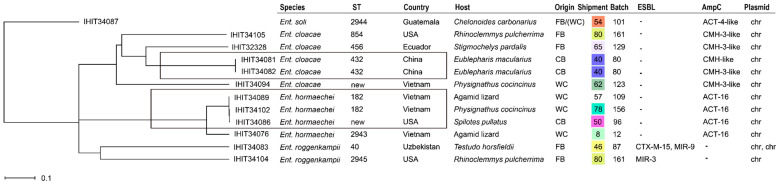
NJ tree for 12 *Enterobacter* species based on pairwise distance calculation of seven alleles used for MLST analysis. chr = chromosomal; nt = not typeable; pm = promoter mutation; CB = captive breed; FB = farm breed; WC = wild-caught.

**Table 1 antibiotics-13-01230-t001:** Data on the number and molecular characteristics of *Enterobacterales* isolates collected in this study.

Approach	Total Number	*E. coli*	*Klebsiella* spp.	*Enterobacter* spp.	*Citrobacter* spp.	Other *Entero-bacterales* *
Isolates from non-selective culturing of 184 samples from 92 shipments	668	*n* = 116/17.4%	*n* = 181/27.1%	*n* = 52/7.8%	*n* = 131/19.6%	*n* = 188/28.1%
Isolates from selective culturing of 668 isolates on cefotaxime agar and subsequent confirmatory test **	79	42	13	12	12	0
Types and numbers of ESBL genes among 79 preselected isolates	39	*bla*_CTX-M-15_ (*n* = 10), *bla*_CTX-M-55_ (*n* = 7), *bla*_CTX-M-27_ (*n* = 3), *bla*_CTX-M-3_ (*n* = 2), *bla*_SHV-12_ (*n* = 2), *bla*_CTX-M-65_ (*n* = 2), *bla*_CTX-M-130_, *bla*_CTX-M-14_, *bla*_CTX-M-1_ (*n* = 1 each)	*bla*_CTX-M-15_ (*n* = 4), *bla*_CTX-M-3_ (*n* = 2), *bla*_SHV-12_ (*n* = 1), *bla*_SHV-42_ (*n* = 1)	*bla*_CTX-M-15_ (*n* = 1)	*bla*_CTX-M-15_ (*n* = 1)	n.t.
Types and numbers of AmpC genes/AmpC promoter mutations among 79 preselected isolates	42/1	*bla*_CMY-2_ (*n* = 9), *bla*_CMY-3_ (*n* = 1), *bla*_DHA-1_ (*n* = 2), *ampC* (−42C > T) (*n* = 1)	*bla*_DHA-1_ (*n* = 4)	*bla*_ACT-16_, *bla*_CMH-like_ (*n* = 4 each), *bla*_ACT-4-like_*, bla*_CMH-9_, *bla*_MIR-3_, *bla*_MIR-9_ (*n* = 1 each)	*bla*_CMY-2_ (*n* = 3)*, bla*_CMY-2-like_ (*n* = 7), *bla*_CMY-4_, *bla*_CMY-101_ (*n* = 1 each), *bla*_CFE-1_, *bla*_DHA-1_ (*n* = 1 each)	n.t.
Multilocus sequence types among 79 isolates positive for ESBL/AmpC/*ampC* promotor mutation		36 different STs: ST48 (*n* = 4), ST58, ST167, ST226 (*n* = 2 each), ST10, ST38, ST90, ST117, ST131, ST155, ST189, ST206, ST218, ST359, ST361, ST398, ST457, ST515, ST540, ST602, ST746, ST757, ST1011, ST1196, ST1408, ST1706, ST1844, ST2705, ST3014, ST3489, ST3580, ST4538, ST4539, ST4553, ST15686, ST15687 (*n* = 1 each)	13 different STs: *K. pneumoniae*: ST16, ST101, ST307, ST432, ST502, ST709, ST1035, ST1087, ST1787, ST1788, ST2054; *K. aerogenes*: ST208, ST719	9 different STs: ST182, ST432 (*n* = 2 each), ST40, ST456, ST854, ST1884, ST2943, ST2944, ST2945 (*n* = 1 each)	10 different STs: ST401 (*n* = 2), ST18, ST96, ST578, ST1210, ST1211, ST1212, ST1213, ST1214, ST1224 (*n* = 1 each)	n.t.

* 69 *Salmonella* spp. isolates and 119 isolates belonging to other *Enterobacterales* isolates; none of the 188 isolates showed growth on cefotaxime selective media. ** Isolates with ESBL or AmpC phenotype plus seven isolates without classic ESBL or AmpC phenotype but reduced sensitivity to the antibiotics used in the confirmatory test.

## Data Availability

The original contributions presented in this study are included in the article and Appendix A. Further inquiries can be directed to the corresponding authors. Raw data, including genome sequencing data, will be made available by the authors on request.

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
