# Peer review of "Phenotypic and Genomic Characterization of ESBL- and AmpC-β-Lactamase-Producing Enterobacterales Isolates from Imported Healthy Reptiles"

_antibiotics, 2024, doi:10.3390/antibiotics13121230_

Round 1
Reviewer 1 Report
Comments and Suggestions for Authors
In this manuscript authors have studied the diversity of ESBLs and AmpC producing Enterobacterales from reptiles. This is an interesting study considering the fact that these reptiles are transported cross border and could be responsible for spreading resistant determinants non-native to their country. However, I have following comments for the authors.
Abstract
1. Line 16-18 : Hypothesis “if harvesting modes, e.g., captive-breeding, harvesting from wild or farmed natural eco-systems might influence microbiota’s antimicrobial resistance (AMR) patterns” has not been discussed in the results and discussion. Rather the manuscript describes the diversity of ESBLs and AmpC in the transported reptiles and the genomic characterization of the ESBL and AmpC producers.
2. Line 25: shows that 79 isolates grew on cefotaxime agar while line 108 says that 99 showed growth on cefotaxime.
Uniformity should be there while describing the number of bacteria would make it easy to follow the text.
Materials and Method
3. Line 567: what does striking resistance pattern means.
4. Line 565-571 is too long and split in two sentence to make it easy to follow.
5. The authors should state clearly that all 79 suspected ESBLs/AmpC producers were Whole genome sequenced (line 577-583).
6. Line 591: It is not clear whether only the diversity of plasmids were determined in section 4.6 using WGS data from Ridom Seqsphere or the genetic locus of ESBLs were determined because next section 4.7 is location of ESBLs/AmpC and line 604-607 implies the same thing.
7. I guess I missed something but What is “see 2.7”. (line 592).
8. Section 4.7 should be renamed to S1-PFGE so that it would easy to follow that 48 isolates were subjected to S1-PFGE for determining the genomic location of ESBLs/AmpC.
9. It should be clearly stated that 48 isolates were subjected to S1-PFGE initially and latter on all 79 were WGS. (rephrase the line 603-604).
10. The rationale behind selecting 48 isolates for S1-PFGE should be stated clearly.
Results
11. Line 90: replace detected with recovered/isolated
12. Line102: Change the sentence to something like Most of the 668 isolates were recovered from samples obtained from USA.
13. Line 108: correct 688 to 668
14. Line 108 -109: what is weak growth. A growth is growth it could be a lawn or a single colony.
15. Section 2.2 is very hard to follow can be rewritten to improve clarity
16. Line 113-114: what is reduced susceptibility pattern, also write the antimicrobials in bracket for there was resistance or intermediate susceptibility.
17. Too much focus have made on describing WT and NWT, whereas the significance of WT and NWT is missing in discussion. This portion could be shorten.
18. The results for ESBLs and AmpC can be separated in different paragraph (from line 166) would be easy to follow.
19. Writing isolates after species name can be avoided through out the manuscript ( for e.g. line178)
20. Line 186 : table S2A
21. Line 189 -191 do not make any sense as they are only number. Rather it can rephrased like x isolates harbored A, B, C genes responsible for resistance to Q,W,E, antibiotics.
22. Line 192 : I would suggest authors should write the results first followed by significance of quinolones and all. Your results should make the 1st impact rather than reader looking for them down somewhere.
23. It would be interesting if authors can add the number of plasmid mediated and/chromosome mediated resistance for quinolone. If there is any region-specific trend.
24. I could not see the results of S1-PFGE (number of plasmids , size). Section 2.4 illustrates the results of diversity of plasmid using WGS. It would be interesting to know how the plasmid content and gene locus varied by S1 PFGE and WGS plasmid profiling.
25. I would suggest authors to use uniformity in writing the numbers as either [n= X, (Y%)] or (X/Y, %).
26. Line 273-274 : clearly state the organism for which ST are mentioned.
Discussion
I suggest shortening the discussion with one paragraph to significance of the study followed by major results and discuss them. At the end one paragraph of limitation and conclusion each.
Comments on the Quality of English Language-
Author Response
In this manuscript authors have studied the diversity of ESBLs and AmpC producing Enterobacterales from reptiles. This is an interesting study considering the fact that these reptiles are transported cross border and could be responsible for spreading resistant determinants non-native to their country. However, I have following comments for the authors.
Authors: We would like to thank Reviewer #1 for this overall positive evaluation. Because we have adopted all linguistic improvements of Reviewer #1 and all three other Reviewers and as the latter did not mention language improvements, we think that language quality is sufficient in the present form.
Abstract
- Line 16-18 : Hypothesis “if harvesting modes, e.g., captive-breeding, harvesting from wild or farmed natural eco-systems might influence microbiota’s antimicrobial resistance (AMR) patterns” has not been discussed in the results and discussion. Rather the manuscript describes the diversity of ESBLs and AmpC in the transported reptiles and the genomic characterization of the ESBL and AmpC producers.
Authors: We have revised accordingly, now stating: “We investigated antimicrobial resistance (AMR) patterns, the diversity of extended spectrum/AmpC β-lactamases (ESBL/AmpC) genes and the genomic organization of the ESBL/AmpC producers.”
- Line 25: shows that 79 isolates grew on cefotaxime agar while line 108 says that 99 showed growth on cefotaxime.
Uniformity should be there while describing the number of bacteria would make it easy to follow the text.
Authors: The statement is nevertheless correct, because of a total of 129 (99 well grown, 30 weakly grown) isolates grown on cefotaxime agar, 41 isolates were suspected ESBL and 31 putative AmpC producers, 7 isolates were negative in the phenotypic test but showed reduced sensitivities to the tested antimicrobials, therefore a total of 79 isolates were further analysed.
Materials and Method
- Line 567: what does striking resistance pattern means.
Authors: The word “striking has been exchanged by “conspicuous”.
- Line 565-571 is too long and split in two sentence to make it easy to follow.
Authors: We have revised accordingly.
- The authors should state clearly that all 79 suspected ESBLs/AmpC producers were Whole genome sequenced (line 577-583).
Authors: This information is already included in section [now] 5.5, stating “Isolates belonging to the species E. coli and to the genera Citrobacter, Enterobacter and Klebsiella and provisionally determined as putative ESBL or AmpC β-lactamase producers were whole genome sequenced.”
- Line 591: It is not clear whether only the diversity of plasmids were determined in section 4.6 using WGS data from Ridom Seqsphere or the genetic locus of ESBLs were determined because next section 4.7 is location of ESBLs/AmpC and line 604-607 implies the same thing.
Authors: For a better understanding, we have switched paragraphs (now) 5.6 and 5.7.
- I guess I missed something but What is “see 2.7”. (line 592).
Authors: Thanks for detecting this error. We have corrected to (see Figures 1-3).
- Section 4.7 should be renamed to S1-PFGE so that it would easy to follow that 48 isolates were subjected to S1-PFGE for determining the genomic location of ESBLs/AmpC.
Authors: The headline of section (now) 5.6 has been revised, now stating “5.6 Location of ESBL and AmpC-β-lactamase genes using S1-PFGE and WGS”.
- It should be clearly stated that 48 isolates were subjected to S1-PFGE initially and latter on all 79 were WGS. (rephrase the line 603-604).
Authors: We have rephrased this paragraph to highlight respective differences of the workflow.
- The rationale behind selecting 48 isolates for S1-PFGE should be stated clearly.
Authors: At the beginning of the study, the isolates that were positive for ESBL/AmpC genes in the PCR were examined using S1 nuclease digestion and subsequent PFGE typing to determine whether the genes were located on a plasmid. As the study progressed, it was decided that all 79 isolates were whole-genome sequenced, among other things to determine the location of the ESBL/AmpC genes.
Results
- Line 90: replace detected with recovered/isolated
- Line102: Change the sentence to something like Most of the 668 isolates were recovered from samples obtained from USA.
- Line 108: correct 688 to 668
- Line 108 -109: what is weak growth. A growth is growth it could be a lawn or a single colony.
Authors: Nos. 11-14 have been revised accordingly.
- Section 2.2 is very hard to follow can be rewritten to improve clarity
Authors: We agree and have revised this section in an effort to improve comprehensibility.
- Line 113-114: what is reduced susceptibility pattern, also write the antimicrobials in bracket for there was resistance or intermediate susceptibility.
Authors: We have revised this paragraph and added specific antimicrobials in the materials and methods part 5.4.
- Too much focus have made on describing WT and NWT, whereas the significance of WT and NWT is missing in discussion. This portion could be shorten.
Authors: We believe that an explanation of WT/NWT is important for the reader to understand a population’s epidemiological resistance patterns, because clinical breakpoint data do not exist for certain antimicrobials/reptile isolates in general. However, we attempted to still address the criticism of Reviewer 1.
- The results for ESBLs and AmpC can be separated in different paragraph (from line 166) would be easy to follow.
Authors: We have addressed this issue by including different sub-headlines to enhance readability.
- Writing isolates after species name can be avoided through out the manuscript ( for e.g. line178)
Authors: We have done accordingly.
- Line 186 : table S2A
Authors: Table S2 has four tabs, which are all relevant to be cited, here. Therefore we have done minor changes to enhance comprehensibility and otherwise ignored to refer only to tab A.
- Line 189 -191 do not make any sense as they are only number. Rather it can rephrased like x isolates harbored A, B, C genes responsible for resistance to Q,W,E, antibiotics.
Authors: We have revised accordingly.
- Line 192 : I would suggest authors should write the results first followed by significance of quinolones and all. Your results should make the 1st impact rather than reader looking for them down somewhere.
Authors: We have revised accordingly.
- It would be interesting if authors can add the number of plasmid mediated and/chromosome mediated resistance for quinolone. If there is any region-specific trend.
Authors: We refer to Supplemental Table S2, where the information of chromosomal mutations in the quinolone resistance-determining region is already included in the columns M and N. We did not explore the location of FQR genes, such as qnr genes. As the tables and figures are already very
- I could not see the results of S1-PFGE (number of plasmids , size). Section 2.4 illustrates the results of diversity of plasmid using WGS. It would be interesting to know how the plasmid content and gene locus varied by S1 PFGE and WGS plasmid profiling.
Authors: We understand that it would be interesting to have this additional information. However, we would like to leave the figures as comprehensive as possible and would not like to add any additional information.
- I would suggest authors to use uniformity in writing the numbers as either [n= X, (Y%)] or (X/Y, %).
Authors: We believe that our data are commonly presented in a harmonized form. Since no standard is required here and because this would necessitate profound revision of the whole manuscript, we would prefer not to realise this suggestion.
- Line 273-274 : clearly state the organism for which ST are mentioned.
Authors: We have clarified this by introducing a colon after E. coli.
Discussion
I suggest shortening the discussion with one paragraph to significance of the study followed by major results and discuss them. At the end one paragraph of limitation and conclusion each.
Authors: A conclusion has been added to the manuscript and several sentences have now been omitted. However, such fundamental criticisms to revise the whole discussion are beyond the possibilities of a minor revision. Because Reviewers #3-4 did not argue to revise the discussion, we would prefer to leave it as an Editor’s decision. Except for the numerous other revisions, we do not see a possibility to address this point in the short given time for revision. Moreover, we still think that the points addressed in the discussion are relevant for a comprehensive understanding of the topic.
Reviewer 2 Report
Comments and Suggestions for Authors
Title: Phenotypic and genomic characterization of ESBL- and AmpC-β-lactamase producing Enterobacterales isolates from imported healthy reptiles
This manuscript is well-written by the authors. I do believe that if they can improve the manuscripts following all comments. It might have a chance to publish in the journal.
Comments
1. Line 18: Methods: A total of 92 shipments with 184 feces……
2. Please write all scientific names and genes in italic.
3. AmpC genes should be written in italic.
4. Line 32-34: Please re-write or modify.
5. Keywords: Enterobacterales, antimicrobial resistance; ESBL, AmpC-β-lactamase; whole genome sequencing
6. I suggest the authors to modify or re-write the introduction. What distinguishes your research between other? please give state of the art of your research
-The first paragraph: describe the importance or situation of imported reptiles (interaction between human and animal) and important bacteria (Enterobacterales) associated with the animals.
-The second paragraph: describe antibiotic resistance and multi-drug resistance in Enterobacterales.
-The third paragraph: describe the information ESBL and ampC genes?
-The fourth paragraph: describe the objective of this study. Why the authors are interested in this study?
7. Do the authors perform a document on ethical approval?
8. For result writing, the authors should not explain whole results. The authors should write as a state of the art.
9. Discussion: Try to compare the results (the author’s hypothesis) with other finding by other researchers.
10. Please delete some introduction and result sentences in the discussion part.
11. Please add a conclusion.
12. Please remove some unnecessary references. Actually, 30-40 references are enough for the research article.
Author Response
This manuscript is well-written by the authors. I do believe that if they can improve the manuscripts following all comments. It might have a chance to publish in the journal.
Authors: We thank Reviever #2 for this pleasant statement.
Comments
- Line 18: Methods: A total of 92 shipments with 184 feces……
Authors: We have revised accordingly.
- Please write all scientific names and genes in italic.
Authors: We agree and have double-checked the content for necessity of italics style. This has been accomplished, also in the Figures 1-3.
- AmpC genes should be written in italic.
Authors: We have revised accordingly.
- Line 32-34: Please re-write or modify.
Authors: We have revised accordingly.
- Keywords: Enterobacterales, antimicrobial resistance; ESBL, AmpC-β-lactamase; whole genome sequencing
Authors: We have revised accordingly, although we think, that keywords should not repeat the content of the title.
- I suggest the authors to modify or re-write the introduction. What distinguishes your research between other? please give state of the art of your research
-The first paragraph: describe the importance or situation of imported reptiles (interaction between human and animal) and important bacteria (Enterobacterales) associated with the animals.
-The second paragraph: describe antibiotic resistance and multi-drug resistance in Enterobacterales.
-The third paragraph: describe the information ESBL and ampC genes?
-The fourth paragraph: describe the objective of this study. Why the authors are interested in this study?
Authors: Again, principal editing of the whole introduction is beyond the possibilities of a minor revision. Because Reviewers #1, 3-4 did not argue to revise the introduction, we would prefer to leave also this issue as an Editor’s decision. Except for the numerous other revisions, we do not see a possibility to address this point in the short given time for revision.
- Do the authors perform a document on ethical approval?
Authors: We have of course checked this issue. Because only fecal and shed skin samples have been used for this study, no ethical approval is necessary.
- For result writing, the authors should not explain whole results. The authors should write as a state of the art.
Authors: To admit, we are not sure, what the reviewers wants us to do here. We do not think, that we have explained the whole results. Moreover, Reviewers #1, 3-4 did not argue to modify this.
- Discussion: Try to compare the results (the author’s hypothesis) with other finding by other researchers.
Authors: We believe that this has already been achieved in the present form, so we did not address this comment.
- Please delete some introduction and result sentences in the discussion part.
Authors: We have done as suggested.
- Please add a conclusion.
Authors: We have done accordingly.
- Please remove some unnecessary references. Actually, 30-40 references are enough for the research article.
Authors: We have critically reviewed the references and removed about 25, but we are still far from reaching only 30-40, which does not seem to be possible in such a complex subject matter.
Reviewer 3 Report
Comments and Suggestions for Authors
General comment
This manuscript is worth publishing. However, the authors will have to work on a few raised issues mentioned below before it gets published.
- Introduction:
L57-8: "... have recently been found in geckos living in close proximity to patients in a hospital in Ghana ..." Provide appropriate citation for this statement.
- Discussion:
L351-2: typo-error. Place a full stop at the end of the sentence.
- Materials and Methods
4.2 Bacterial Isolates and DNA Extraction:
L527: " ... bovine serum..." please provide either the catalogue number or detailed origin (name of the manufacturer\processor, town and country) of the medium.
L530: "... glycerol..." please provide either the catalogue number or detailed origin (name of the manufacturer\processor, town and country) of the medium.
4.6 Determination of AMR genes, chromosomal mutations and plasmid types
L590: "Furthermore …” repetition, please use an alternative adverb instead this one.
- Reference
L774-5: This citation is not meant for this work. Please get a translated version of it.
Author Response
This manuscript is worth publishing. However, the authors will have to work on a few raised issues mentioned below before it gets published.
Authors: Thanks to Reviewer #3 for this overall positive evaluation.
Introduction:
L57-8: "... have recently been found in geckos living in close proximity to patients in a hospital in Ghana ..." Provide appropriate citation for this statement.
Authors: The citation has been updated and is now active.
Discussion:
L351-2: typo-error. Place a full stop at the end of the sentence.
Authors: We have done accordingly.
Materials and Methods
4.2 Bacterial Isolates and DNA Extraction:
L527: " ... bovine serum..." please provide either the catalogue number or detailed origin (name of the manufacturer\processor, town and country) of the medium.
L530: "... glycerol..." please provide either the catalogue number or detailed origin (name of the manufacturer\processor, town and country) of the medium.
4.6 Determination of AMR genes, chromosomal mutations and plasmid types
L590: "Furthermore …” repetition, please use an alternative adverb instead this one.
Authors: We have revised Materials and Methods accordingly.
Reference
L774-5: This citation is not meant for this work. Please get a translated version of it.
Authors: This reference is important to prove principal import numbers, which is scarce in literature. Although this citation is not from a peer-reviewed journal, but from a serious source (state laboratory annual report) and no translated work is available, we decided to leave this citation.
Reviewer 4 Report
Comments and Suggestions for Authors
Major points
- Table 1 and Results 2.1 – some of the results presented here would be easier to read in the form of a table/ graph (e.g., the different bacterial species, countries of origin etc).
- Same as above for section Results 2.2, 2.3
- Table 1 and Section 2.3 – add the total number of resistance genes in the table – it would make the parallel with the text more accessible
- Abbreviations such as Ec, Kp etc decrease the readability of the manuscript. Please maintain the usually employed bacterial nomenclature (e.g., E. coli etc)
- Similarly to the points mentioned above, the discussion section is well written; however, it is somewhat challenging to follow. One possible solution for this would be to introduce a table in the discussion section to compare the identified plasmids/genes etc with the data available from the literature.
- Line 565 – Why were CLSI and not EUCAST guidelines followed?
- The manuscript contains no conclusions section. If the paragraphs contained between lines 474-491 are intended to be the conclusions, please note this section as such and rewrite it in a way that summarises the findings of the study
Minor points
- Lines 546-550 – a reference would be helpful here regarding the limitations of Enterobacter/Citrobacter identification on MALDI-TOF MS
- The titles for each Excel sheet in the supplementary File are not clearly formatted
- Some minor aspects regarding the English language require improvements
- Line 434 – replace “ A Mexican study “ with a study from Mexico or similar
Author Response
Major points
- Table 1 and Results 2.1 – some of the results presented here would be easier to read in the form of a table/ graph (e.g., the different bacterial species, countries of origin etc).
- Same as above for section Results 2.2, 2.3
- Table 1 and Section 2.3 – add the total number of resistance genes in the table – it would make the parallel with the text more accessible
Authors: We do not fully agree with the first two comments, because all data can already additionally be found in respective Figures, Tables and Supplementary Tables. Especially Suppl. Table S2 (excel file) can be individually adjusted and used for this purpose. We have also tried several versions of Table 1 before and although we agree that more data would be nice, here, the space is too limited. Otherwise, a violation of the author guidelines cannot be avoided with respect to the table style.
- Abbreviations such as Ec, Kp etc decrease the readability of the manuscript. Please maintain the usually employed bacterial nomenclature (e.g., E. coli etc)
Authors: We have revised accordingly.
- Similarly to the points mentioned above, the discussion section is well written; however, it is somewhat challenging to follow. One possible solution for this would be to introduce a table in the discussion section to compare the identified plasmids/genes etc with the data available from the literature.
Authors: We thank Reviewer #4 for this suggestion and have now included such a Table (Table 2) in the main text.
- Line 565 – Why were CLSI and not EUCAST guidelines followed?
Authors: We used CLSI guidelines for the detection of ESBL/AmpC producers and switched to EUCAST for MIC interpretations, as this was more suitable for the isolate collection from reptiles.
- The manuscript contains no conclusions section. If the paragraphs contained between lines 474-491 are intended to be the conclusions, please note this section as such and rewrite it in a way that summarises the findings of the study
Authors: We have revised accordingly and added a conclusion section.
Minor points
- Lines 546-550 – a reference would be helpful here regarding the limitations of Enterobacter/Citrobacter identification on MALDI-TOF MS
Authors: We have revised accordingly.
- The titles for each Excel sheet in the supplementary File are not clearly formatted
Authors: We have revised accordingly.
- Some minor aspects regarding the English language require improvements
- Line 434 – replace “ A Mexican study “ with a study from Mexico or similar
Authors: We have revised accordingly.